# Harnessing Multi-role Capabilities of Large Language Models for Open-domain Question Answering

## ABSTRACT

Open-domain question answering (ODQA) stands as a pivotal research spotlight in web mining. Existing ODQA methods follows two main paradigms to collect evidence: 1) The *retrieve-then-read* retrieves pertinent documents from an external corpus; and 2) the *generate-then-read* paradigm, which employs large language models (LLMs) to generate relevant documents. Despite both paradigms have their own advantages, a single paradigm cannot take into account multifaceted requirements for evidence. To this end, we propose LLMQA, a generalized framework that formulates the ODQA process into three basic steps: query expansion, document selection, and answer generation, which is a novel paradigm combining the superiority of retrieval-based and generation-based evidence. Existing research has verified that LLMs can exhibit their excellent capabilities to accomplish various types of tasks. Therefore, in contrast to previously utilizing specialized models to complete each individual module of ODQA, we instruct LLMs to play multiple roles as generators, rerankers, and evaluators in our unified framework, and integrate them to collaborate each other to jointly enhance the performance of ODQA task. Furthermore, we introduce a novel prompt optimization algorithm to refine the role-play prompts and steer LLMs towards producing higher-quality evidence and more accurate answers. We conduct extensive experiments on three widely used benchmarks: NQ, WebQ, and TriviaQA. Experimental results demonstrate that our LLMQA can achieve the best performance in terms of both answer accuracy and evidence quality, showcasing its potential for advancing ODQA research and applications.

## CCS CONCEPTS

• **Information systems** → **Data mining**; • **Computing methodologies** → **Natural language generation**; **Natural language processing**;

## KEYWORDS

Question answering, Large language models, Prompt optimization

**ACM Reference Format:**
Anonymous Author(s). 2018. Harnessing Multi-role Capabilities of Large Language Models for Open-domain Question Answering. In *Proceedings of Make sure to enter the correct conference title from your rights confirmation emai (Conference acronym 'XX).* ACM, New York, NY, USA, 10 pages. https://doi.org/XXXXXXX.XXXXXXX

## 1 INTRODUCTION

In the interdisciplinary fields of information retrieval and web mining, open-domain question answering (ODQA) stands as a pivotal research spotlight. It is intrinsically a knowledge-intensive task, which predominantly focuses on answering factoid questions, eschewing the limitations of a pre-specified domain, thus enhancing its applicability in a diverse range of web scenarios [5, 26, 28].

Current ODQA methods commonly follows two main paradigms to collect evidence in preparation for answering questions: 1) The *retrieve-then-read* paradigm retrieves a set of pertinent evidence documents from an external corpus, and then generates the answer based on them [13, 15]. Since retrieval models often rely on existing documents in well-curated corpora like Wikipedia, they can provide highly factual and accurate information for answering the factual question; 2) The *generate-then-read* paradigm directly employs language models to generate evidence for producing the final answer [45]. These generated virtual documents may diversify the sources of evidence, enhancing answer coverage for the question.

Despite both paradigms have their own advantages, a single paradigm cannot take into account multifaceted requirements for evidence. An intuitive idea is to reasonably integrate the advantages of these two paradigms so that the collected evidence obtained contains both factual reliability and diversity. To this end, we propose LLMQA, a novel generalized framework that combines the strengths of retrieval-based and evidence-based evidence for ODQA. Specifically, we formulate the generation process of ODQA into three fundamental steps: 1) **Query expansion** involves expanding the given question and generating background passages or explanations, serving as generated-based evidence to enrich the context; 2) **Document selection** integrates retrieval-based evidence by reranking the retrieved documents, increasing their relevance to the answer and the likelihood of covering the answer; 3) **Answer generation** proceed to generate the final answer based on comprehension of the question and evidence.

In order to implement each step of ODQA, previous methods carefully train specialized models on each individual module to obtain various paradigms of evidences and final answers. Limited by the inherent capabilities of these models, jointly optimizing each module to improve overall performance remains challenging. Existing works have demonstrated that large language models (LLMs) can exhibit their excellent capabilities to accomplish various types of tasks [4]. Specific to the ODQA task, it mainly needs to integrate text generation [9, 30, 41, 45], document ranking [9, 22, 23], and candidate evaluation [2, 47], the three aspects of capabilities of LLMs into each module. Therefore, we aim to instruct LLMs to play the three roles of generators, rerankers, and evaluators respectively under our proposed unified framework. To fully explore the potential of these roles and improve model performance, we closely coordinate individual roles and make them collaborate each other to complete each module of ODQA. As shown in Figure 1: 1) The

*generator* is responsible for employing LLMs to expand the query for more informative description, which can provide comprehensive and pertinent information for the answer generation; 2) The *reranker* plays an important role in document selection by reranking and prioritizing the retrieved documents to distill more valid and relevant documents as evidence; 3) The ***evaluator*** engages in interacting with the generator and the reranker through evaluative feedback by scoring their candidates, instructing them to generate more refined correspondences.

Furthermore, the distinct roles played by LLMs rely on unique prompts that describe task definitions and guide their behaviors. Therefore, the performance of different role-play LLMs is highly dependent on the quality of the prompts. The precision of evidence generated or reranked by LLMs is also sensitive to the prompts used. To better automatically design prompts, we present a novel prompt optimization algorithm to enhance the effectiveness of LLMs playing different roles in our unified framework. During the generation process of ODQA, we consider the evidences (*i.e.,* query expansion and selected documents) as latent variables, and creatively leverage variational inference to learn their distributions, and then optimize crucial role-play prompts to steer LLMs towards producing higher-quality evidence and more precise answers.

We conduct comprehensive experiments on three widely used ODQA benchmarks: NQ, WebQ, and TriviaQA. Experimental results show that our LLMQA advances the state-of-the-art performance on both answer accuracy and evidence quality. Compared with previous baseline models, our LLMQA achieve remarkable improvement on EM scores (4.0@TriviaQA, 2.7@WebQ, 3.1@NQ), demonstrating the effectiveness of multi role-play LLMs for ODQA. Our proposed role of generator for query expansion can achieve up to 73%,76% and 87% recall for the target answer in generated expansions. The role of reranker increased answer coverage about 8.1% averaged on three datasets. Our ablation results not only suggest that each role-play LLMs and prompt optimization can contribute to the improvement, but also verifies that multi-roles of LLMs can cooperate with each other.

Overall, our main contributions can be summarized as follows:
- We propose LLMQA, a generalized framework model to formulate the generation process of ODQA, which is a novel paradigm to combine the strengths of retrieval-based and generation-based evidence.
- We effectively instruct LLMs to play three roles of generators, rerankers, and evaluators respectively and integrate their collaborative interactions under our proposed unified framework,
- We present a novel prompt optimization algorithm to guide LLMs in producing higher-quality evidence and more precise answers. Extensive experimental results verifies LLMQA advances state-of-the-art performance in terms of both answer accuracy and evidence quality.

## 2 RELATED WORK

### 2.1 Open-Domain Question Answering

ODQA requires external knowledge to generate answers, thus it has become a common and effective benchmark to measure abilities of natural language comprehension and generation in web mining [26]. For collecting the related documents as evidence, existing methods

can be categorized into two main paradigms: *retrieve-then-read* and *generate-then-read.*

**Retrieve-then-read paradigm.** Pioneered by [5], most recent approaches consist of two main modules: *retriever* and *reader.* The retriever module firstly retrieves a small set of documents relevant to the given question from an external knowledge base. The reader module then comprehends on both question and retrieved documents and generates the corresponding answer. One branch of the recent approaches focus on improving the *retriever.* Sparse retrieval with inverted indexes (*e.g.,* TF-IDF or BM25) is generally used in traditional approaches [36]. Dense retrieval using language models such as ORQA [17], DPR [15], RocketQA [31], ColBertQA [16] and ART [37] is the dominant method nowadays. The other branch focuses on enhancing the comprehension ability of *reader* to generate more appropriate and accurate answers [6, 13]. Especially with the development of LLMs, most *readers* are adopted from fine-tuned T5 [33] or InstructGPT [27].

**Generate-then-read paradigm.** Previous works have demonstrated that the knowledge preserved in LLMs can serve as a "*generative retriever*" [29, 32, 35]. Although many existing approaches adopt LLMs in ODQA, they cannot fully harness the generation capability of LLMs [14, 30, 41, 42]. Unlike conventional methods that retrieve documents from external sources, GenRead is the first to explore the potential of generate-then-read pipeline in ODQA, which instructs an LLM to generate a set of documents base on clusters of question-document pairs and the given question [45]. Then these generated documents and the question are fed into LLM together to produce the final answer.

Considering the limitations of a single paradigm, we propose to seamlessly integrate retrieval- and generation-based evidence in the ODQA generation process, by effectively harnessing the capabilities of LLMs.

### 2.2 Capabilities of LLMs

With recent notable enhancements in model scales [8, 27], LLMs have showcased impressive capabilities in text generation, ranking and evaluation.

**Generation capability of LLMs.** Recent studies have highlighted the superior text generation capability of LLMs in few-shot and zero-shot scenarios [3, 4, 8, 48]. Previous works have demonstrated that the knowledge stored in LLMs could be retrieved during inference [29, 35]. Building upon this, some studies directly prompt LLMs to generate answers conditioned on the question in ODQA [14, 30, 41, 42]. Other approaches utilize the generation capability of LLMs to expand the query or enrich the context [9, 25, 45].

**Ranking capability of LLMs.** Previous works show that compared to few-shot information extraction, LLMs are better at reranking for hard examples. Ma et al. propose a *filter-then-rerank* paradigm, which utilize LLMs to rerank the candidates filtered by smaller language models to generate the final response. Chuang et al. apply LLMs to rerank a diverse set of expanded queries and select those leading to better results. Ma et al. replace pointwise reranking with listwise reranking to reorder the list of documents base on the relevance to the query.

**Evaluation capability of LLMs.** LLMs are chosen as evaluators due to their robust comprehension and reasoning capabilities. [4, 7]. Weng et al. leverage the self-verification capability of LLMs for better reasoning. Shinn et al. use self-reflective feedback as a semantic gradient providing with a concrete direction to learn from prior mistakes. Madaan et al. present an iterative self-refinement algorithm that alternates between feedback and refinement. Additionally, LLMs are also used to evaluate attribution between generated answer and references [2, 47].

In this paper, we aim to effectively integrate multi-role capabilities of LLMs to enhance the overall performance on ODQA.

## 2.3 Prompt Optimization

Previous works have emphasized that subtle differences in prompts could lead to tremendous performance degradation in generated results [11, 19, 49]. Consequently, prompt optimization has attracted great attention in recent years, with two primary approaches: manual design [34] or automatic generation [38]. Gradient-based prompt tuning can optimize prompts embedding in a continuous space [20, 21]. In contrast, discrete prompt optimization has been extensively studied including prompt scoring [10], prompt generation [11] and prompt paraphrasing [46]. Recently, Zhou et al. propose APE for automatic prompt optimization by iteratively selecting prompt candidates to maximize the potential score functions. DLN [40] steps further by viewing LLMs as language layers and prompts as learnable parameters.

Inspired by these methods, we present a novel prompt optimization algorithm to refine essential prompts for query expansion, document reranking, and answer generation, enabling LLMs to produce better evidences and answers.

## 3 METHOD

### 3.1 Task Formulation

ODQA places a central emphasis on acquiring pertinent evidence to enhance the reliability and precision of answers to a given question. Previous methods collect evidences by retrieving or generating relevance background passages or explanations to facilitate accurate answer identification [5, 15, 27]. Expanding upon this concept, we formulate the generation process of ODQA as the following three fundamental steps: 1) **Query expansion**: We commence with the input question, designated as query $q$. To enrich the context and improve document selection and answer generation, we utilize knowledge stored in language models to generate additional background information, denoted as query expansion $e$; 2) **Document selection**: Leveraging both the query $q$ and its expansion $e$, we initially retrieve the top-$n$ documents that are relevant to answering the question as candidates. Subsequently, we compare within these candidates to prioritize those documents most likely to contain the answer. Based on this criterion, we rerank these $n$ candidates and retain top-$k$ documents, represented as $d$, which collectively constitute the evidence in conjunction with query expansion $(e, d)$; 3) **Answer generation**: Based on the query $q$ and the derived evidence $(e, d)$, we proceed to generate the final answer $a$ in response to the question with a reader model.

Furthermore, this generation process can be effectively formulated using a Bayesian graphical model that aligns closely with

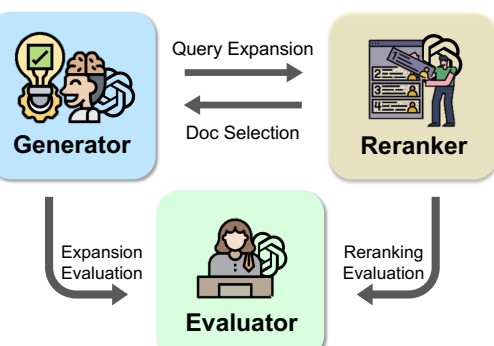

**Figure 1: Collaborative interactions of multiple LLM roles.**

the three aforementioned steps, parameterized by the following probability distribution:

$$P(a|q) = \sum_e \sum_d P(e|q)P(d|q, e)P(a|q, e, d), \quad (1)$$

where we consider the evidence $(e, d)$ as latent variables, which require to be optimized by maximizing this marginal likelihood. Consequently, the acquisition of the most appropriate evidence for question answering becomes a critical aspect of this task. Considering the prominent performance of LLMs on various tasks including text generation, ranking, and evaluation, we propose to harness LLMs in multiple roles that cooperate with each other and seamlessly integrate into the ODQA generation process. The framework overview of our LLMQA is shown in Figure 2. In the subsequent sections, we will introduce in detail how to leverage the multi-role capabilities of LLMs to enhance the ODQA task.

### 3.2 Query Expansion

Generally, the questions posed in ODQA datasets are brief and concise, indicating that relying solely on the question itself as query can lead to a substantial challenge: inadequate query context makes it difficult to support accurate document selection and answer generation. To address this challenge, we add a pivotal step known as query expansion that aims to enrich the original question with a broader context. The generated expansions are mainly used to analyze the key points required to answer a given question and provide more sufficient background information for subsequent steps. In this process, we instruct an LLM to play the role of generator leveraging its powerful context understanding and text generation capabilities. Specifically, we employ an LLM-based expansion generator $G_e$ to facilitate the query expansion step. Given a question $q$, its query expansion $e$ can be generated by

$$e = G_e(q; \theta_e), \quad (2)$$

where $\theta_e$ represents the prompt to instruct the query expansion.

### 3.3 Document Selection

In addition to the query expansion, relevant documents are more commonly used as evidences to include accurate answers of the question. To identify the most appropriate documents, we divide this document selection process into two distinct stages:

**1) Coarse-grained retrieval of top-$n$ documents:** we first retrieve a set of top-$n$ documents that are potentially relevant to

**Figure 2: The overview of our LLMQA. Three different role-play LLMs execute five main steps: (a) generate query expansion according to the question by *generator*; (b) select the best query expansion by *evaluator*; (c) rerank the top-$k$ documents according to the question and generated expansion by *reranker*; (d) select the best rerank documents by *evaluator*; (e) generate answer according to the question, generated expansion and reranked documents by *generator*. A more detailed insight into sliding window reranking: select top 2 documents from top 5 retrieved candidates with window size $w = 3$, step $l = 1$.**

the given question by employing established information retrieval techniques such as DPR [15] or BM25 [5]. These retrieval methods provide an initial score for each candidate to describe the relevance between documents and questions. However, such methods may not always capture nuanced semantic relationships between the query and documents, leading to false positives or irrelevant documents in the initial set.

**2) Fine-grained reranking of top-$k$ documents from $n$ candidates:** we proceed with the reranking of documents to ensure that those more likely to contain the answer are prioritized. This stage involves comparing the documents to determine which ones exhibit higher quality and relevance to the query. Inspired by LLM-based ranking approaches [9, 22, 23], We instruct LLM to play the role of document reranker $R_d$ for further screening out top-$k$ ($k < n$) documents from the initial pool of $n$ candidates. Considering the limitation on input tokens for LLMs, we iteratively rerank a subset of documents each time and complete the reranking of all candidates through a sliding window. Specifically, we set the window size to $w$ and the step size to $l$. We start from the last position of the initially sorted documents. In each iteration, we focus on comparing $w$ documents within the sliding window, and reorder the documents based on their likelihood of containing the answer. With the sliding window moving forward by $l$ steps, thus the top $w - l$ reranked documents in the original window are reserved and $l$ new documents are added, then the next $w$ documents can be reordered. This iterative process continues until the sliding window reaches the front, and we consider the first $k = w - l$ documents as the final evidence documents $d$. Overall, this document selection process can be simplified as:

$$d = R_d(q, e; \theta_d), \tag{3}$$

where $\theta_d$ denotes the prompt for $R_d$ to ensure that documents are ranked in alignment with the desired relevance and quality.

## 3.4 Answer Generation

Based on the query $q$, and the evidence $(e, d)$, the final step in ODQA is to generate the final answer with the integration and comprehension of pertinent information within the evidence. The evidence can encompass essential information that directly provides the answer to the question, or it may comprise an analysis and explanation necessary for formulating the answer. Consequently, the central objective of answer generation is to employ a reader model for the systematic extraction and comprehension of valuable insights from the evidence context. We utilize an LLM-based reader $G_a$ to generate a precise and dependable response as the predicted answer to the question, and formulate this process as:

$$a = G_a(q, e, d; \theta_a), \tag{4}$$

where $\theta_a$ indicates the prompt for answer generation to ensure that the generated answer can align with the context and requirements of the original question and its evidence.

## 3.5 Evaluators for Generation and Reranking

As shown in Figure 1, evaluators also play a crucial role in query expansion and document reranking, engaging in a dynamic interaction with both the generator and reranker. Leveraging the advanced capabilities of LLMs to evaluate text quality under specific standards, we can instruct LLMs to play the role of evaluators to assess the performance of the generator and the reranker. The primary objective of evaluators is to assign quality scores to multiple candidates generated by the generator and reranker. These scores reflect the likelihood that each candidate is appropriate and accurate for

---

**Algorithm 1** Training Process for Prompt Optimization.

---

**Input**: Training data: $X_{tra}$, LLM-based roles: $G_e, R_d, G_a, S_e, S_r, S_a$,
backward updating functions: $U_e, U_d, U_a$.

**Parameters**: $\theta_e, \theta_d, \theta_a$.

1: **for** $(q, a)$ in $X_{tra}$ **do**
2:     Generate the prior $\widetilde{e} = G_e(q; \theta_e)$ by Expansion Generator
3:     Generate the prior $\widetilde{d} = R_d(q, \widetilde{e}; \theta_d)$ by Document Reranker
4:     Generate the prior $\widetilde{a} = G_a(q, \widetilde{e}, \widetilde{d}; \theta_a)$ by Answer Generator
5:     Sample $n$ posterior $\widehat{d}_1, \widehat{d}_2, \cdots, \widehat{d}_n$ from $R_d(q, \widetilde{e}, \widetilde{d}, a; \phi_d)$
6:     Score $s_{d_i} = S_r(R_d(q, \widetilde{e}), \widehat{d}_i)$ and $S_a(G_a(q, \widetilde{e}, \widehat{d}_i), a)$ for each $\widehat{d}_i$
7:     Calculate posterior score $v_{d_i} = s_{d_i} * s_{a_i}$
8:     Select the best posterior $\widehat{d}_* = \text{argmax}_i \{v_{d_i}\}$
9:     Sample $m$ posterior $\widehat{e}_1, \widehat{e}_2, \cdots, \widehat{e}_m$ from $G_e(q, \widetilde{e}, \widehat{d}_*, a; \phi_e)$
10:     Score $s_{e_j} = S_e(G_e(q), \widehat{e}_j)$, $s_{d_j} = S_r(R_d(q, \widehat{e}_j), \widehat{d}_*)$, and $s_{a_j} = S_r(G_a(q, \widehat{e}_j, \widehat{d}_*), a)$ for each $\widehat{e}_j$
11:     Calculate posterior score $v_{e_j} = s_{e_j} * s_{d_j} * s_{a_j}$
12:     Select the best posterior $\widehat{e}_* = \text{argmax}_j \{v_{e_j}\}$
13:     Sample $K$ candidates $\widehat{\theta}_{a_k} = U_a(q, \widetilde{e}, \widetilde{d}, \widetilde{a}, a)$ for $\theta_a$
14:     Select $\widehat{\theta}_{a_*} = \text{argmax}_k \sum_{i=1}^{n} \sum_{j=1}^{m} v_{e_j} v_{d_i} \log S_e(G_a(q, \widehat{e}_j, \widehat{d}_i; \widehat{\theta}_{a_k}), a)$
15:     Sample $K$ candidates $\widehat{\theta}_{d_k} = U_d(q, \widetilde{e}, \widetilde{d}, \widehat{d}_*)$ for $\theta_d$
16:     Select $\widehat{\theta}_{d_*} = \text{argmax}_k \sum_{i=1}^{n} \sum_{j=1}^{m} v_{e_j} v_{d_i} \log S_r(R_d(q, \widehat{e}_j; \widehat{\theta}_{d_k}), \widehat{d}_i)$
17:     Sample $K$ candidates $\widehat{\theta}_{e_k} = U_e(q, \widetilde{e}, \widehat{e}_*)$ for $\theta_e$
18:     Select $\widehat{\theta}_{e_*} = \text{argmax}_k \sum_{j=1}^{m} v_{e_j} \log S_e(G_e(q; \widehat{\theta}_{e_k}), \widehat{e}_j)$
19:     Update parameters: $\theta_a \leftarrow \widehat{\theta}_{a_*}, \ \theta_d \leftarrow \widehat{\theta}_{d_*}, \ \theta_e \leftarrow \widehat{\theta}_{e_*}$
20: **end for**
21: **return** $\theta_e, \theta_d, \theta_a$.

---

specific conditions or requirements. For a given question, we employ the expansion evaluator $S_e$ to individually score each candidate expansion ranging from 0 to 1, which is used to assess the degree of their relevance and logical consistency. Similarly, we use the reranking evaluator $S_r$ to score different top-$k$ reranking candidates generated by the reranker $R_d$, assessing the contribution of each ranking result to answering the question. The scoring process of evaluators $S_e$ and $S_r$ can be formulated as:

$$s_{e_j} = S_e(G_e(q), e_j), \tag{5}$$

$$s_{d_j} = S_r(R_d(q, e), d_j), \tag{6}$$

where $s_{e_j}$ and $s_{d_j}$ represent the scores assigned to the $j$-th candidate of generated query expansion and reranked documents. These scores serve as critical metrics for evaluating the performance of the generator and reranker and further promoting overall generation and ranking capabilities of LLMs.

## 3.6 Prompt Optimization

The role-play performance of generator and reranker still heavily relies on the prompt design in each ODQA generation process. Therefore, we explore how to design better role-play prompts or expansion generation $\theta_e$, document reranking $\theta_d$, and answer generation $\theta_a$ to fully exploit the potential of LLMs. We propose a novel

algorithm to enable prompt optimization under the unique graphical model structure of ODQA. Throughout the ODQA generation process, we do not require the LLM parameters, but instead treat three natural language prompts as learnable parameters. In Equation (1), the distributions of latent variables $e$ and $d$ are determined by these prompts and need to be approximated by probabilistic inference techniques. To ensure consistency with the graphical model, we propose to use variational inference to learn the hidden distributions and optimize prompts. We denote the prior distribution as $P_\theta$ and the posterior distribution as $P_\phi$, and the original log-likelihood could be bounded by the following ELBO:

$$\log P(a|q)$$
$$\geq \sum_e \sum_d P_{\phi_e}(e|q, a) P_{\phi_d}(d|q, e, a) \log \frac{P_{\theta_e}(e|q) P_{\theta_d}(d|q, e) P_{\theta_a}(a|q, e, d)}{P_{\phi_e}(e|q, a) P_{\phi_d}(d|q, e, a)}. \tag{7}$$

As shown in Algorithm 1, for the question $q$, we use predefined $G_e$, $R_d$ and $G_a$ to sequentially simulate the priors $P_{\phi_e}, P_{\phi_d}$, and $P_{\phi_a}$, and generate the query expansion $\widetilde{e}$, the reranked documents $\widetilde{d}$ and the predicted answer $\widetilde{a}$ during forward inference. Next, to approximate the posteriors $P_{\phi_e}$ and $P_{\phi_d}$, we consider the following two aspects: 1) We add the ground-truth target as an additional condition to estimate the posteriors; 2) We sample several posterior candidates near the prior to ensure low Kullback–Leibler (KL) divergence between them in the space of discrete texts. Specifically, we sample $n$ posterior candidates $\widehat{d}_1, \widehat{d}_2, \ldots, \widehat{d}_n$ by replacing the last document in $\widetilde{d}$ with the $(k+1)$-th, $(k+2)$-th, ..., $(k+n)$-th document, respectively. Then we use evaluators $S_r$ and $S_e$ to score each posterior candidate for estimating $P_{\phi_d}$. The best posterior $\widehat{d}_*$ among these candidates is selected as the current "ground-truth" reranking documents. This process can be repeated similarly for the query expansion to obtain the best posterior $\widehat{e}_*$.

Subsequently, we define a backward process to update prompts. For the answer generation prompt $\theta_a$, we sample $K$ candidates near it using a updating function $U_a(q, \widetilde{e}, \widetilde{d}, \widetilde{a}, a)$, to guide prompts to update in a direction that brings the predicted answer $\widetilde{a}$ closer to the actual answer $a$. Then all the previous posterior candidates are used to estimate ELBO, and the best $\widehat{\theta}_{a_*}$ to maximize ELBO can be selected as the refined prompt for answer generation. Similar processes are introduced to refine prompts $\theta_d$ and $\theta_e$, while updating functions $U_d$ and $U_e$ are used to guide the directions to refine document reranking and query expansion.

## 4 EXPERIMENTS

### 4.1 Experimental Setup

**Datasets.** We select three widely used ODQA benchmarks to evaluate the model performance of baselines and our LLMQA: 1) **WebQ** (WebQuestions) is a dataset that consists of questions obtained using the Google Suggest API, with the answers being entities from Freebase. 2) **NQ** (Natural Questions) is a dataset generated from real Google search queries, and the answers are spans within Wikipedia articles. 3) **TriviaQA** is a collection of trivia questions sourced from trivia and quiz-league websites.

**Baselines.** To verify the effectiveness of our method, we compare LLMQA with the following two main types of baselines: 1) Baselines without LLMs: **BM25+Bert** [17] combines sparse retrieval methods with BERT for text representations. **REALM** [12] retrieves relevant documents from a knowledge corpus and incorporate them into

**Table 1: Comparison results on TriviaQA, WebQ, and NQ datasets. Our EM scores are given by the mean of 10 rounds of bootstrapping sampling, with bold numbers indicating $p$-values below 0.01 under a significance test.**

| Method | #Reader parameters | #Documents | TriviaQA | WebQ | NQ |
|---|---|---|---|---|---|
| *baselines without LLMs; † was reported by paper, ∗ was reproduced by our* | | | | | |
| BM25+Bert[†] | 220M | 5 | 47.1 | 21.3 | 26.5 |
| REALM[†] | 330M | 5 | - | 40.7 | 40.4 |
| DPR[†] | 110M | 100 | 56.8 | 41.1 | 41.5 |
| RAG[†] | 400M | 10 | 56.1 | 45.2 | 44.5 |
| FiD-l[†] | 770M | 10 | 61.9 | 48.1 | 46.7 |
| FiD-xl[†] | 3B | 10 | 66.3 | 50.8 | 50.1 |
| *Baselines employing LLMs as generators; † was reported by paper, ∗ was reproduced by ours.* | | | | | |
| GenRead (FiD-l) (sampling)[†] | 770M | 10 | 67.8 | 51.5 | 40.3 |
| GenRead (FiD-l) (clustering)[†] | 770M | 10 | 70.2 | 53.5 | 43.5 |
| GenRead (FiD-xl)) (sampling)[†] | 3B | 10 | 69.6 | 52.6 | 42.6 |
| GenRead (FiD-l) (clustering)[†] | 3B | 10 | 71.6 | 54.4 | 45.6 |
| EAR+FiD-l[†] | 770M | 100 | 71.2 | - | 51.4 |
| EAR+FiD-xl[∗] | 3B | 100 | 72.9 | - | 53.8 |
| LLMQA | 3B | 5 | **76.9** | 56.2 | 56.9 |
| LLMQA | 3B | 10 | 76.6 | **57.1** | **57.5** |

the training process of the language model. **DPR** [15] utilizes a dense encoder to encode text passages and questions and retrieves relevant passages based on vector similarity. **RAG** [18] utilizes retrieval to augment generation techniques to enhance the ODQA tasks. **FiD** [43] follows the classic retrieve-then-read paradigm with reader sizes of 770M and 3B. 2) Baselines employing LLMs as generators: **GenRead** [45] propose a clustering-based method to use LLMs to generate diverse documents. **EAR** [9] improves evidence quality through query re-ranking for enhanced expansion.

**Evaluation Metrics.** Mainstream ODQA methods evaluate answer accuracy using the Exact Match (EM) score [50], which compares the predicted answer $\widetilde{a}$ to each ground-truth answer $a$ in the answer list, to determine if they match. Additionally, the recall score serves as an important metric for assessing the quality of evidence. These two metrics are given by:

$$EM = \frac{\sum_{\widetilde{a}, a \in D} exact\_match(\widetilde{a}, a)}{|D|} \quad (8)$$

$$recall = \frac{\sum_{docs, a \in D} answer\_hit(docs, a)}{|D|} \quad (9)$$

where $D$ is the dataset, $\widetilde{a}$ is the predict answer, $a$ is the ground truth answer, $docs$ is the reference documents.

**Implementation Details.** In our proposed approach, we take advantage of the multi-role capability of LLMs. As for the query expansion, we use *gpt-3.5-turbo-16k* as generator by directly accessing to API (temperature=0.7,n=10). As for documents selection, we first retrieved top 100 documents using DPR [15]. To select top 10 reranked documents, we implemented sliding window reranking and set the window size $w = 20$, step $l = 10$. We get the rerank result in the window by accessing to *gpt-3.5-turbo-16k* (temperature=0.7) as well. As for answer generation, we followed GenRead [45] adapting FiD-xl (3B) as reader model and finetune it for 10000 steps with $lr$ set to 3e-5. As for prompt optimization, we refer to P3 [1] along

with carefully designed role-play instructions to initialize crucial prompts. As for evaluator used in prompt optimization, we use *text-embedding-ada-002* from OpenAI by requesting for embeddings to estimate the posterior probability.

## 4.2 Overall Performance

The overall performance of the experiment is shown in Table 1. Compared with the baselines without LLMs, our proposed LLMQA exhibited a notable improvement over three datasets (10.3@TriviaQA, 6.3@WebQ, 7.4@NQ), which strongly demonstrated the effectiveness of the LLM on the ODQA, indicating that the effect of model scale on the final results is remarkable. Our LLMQA surpassed **FiD-xl** by 8 on average of three datasets, even though the documents we used are less than it. Thus, different role-play LLMs can be competent with previously specifically designed models.

Compared with the baselines employing LLMs as generators, our LLMQA also achieved considerable performance improvement. Both **GenRead** and **EAR+FiD** utilize the generation capability of LLMs to generate documents or query expansions. The enhancement of our approach primarily leverages the collaboration between multiple role-playing LLMs. In addition to the query expansion used in our approach, we also adapted LLMs to rerank the retrieved documents. The remarkable improvement fully demonstrated that LLMs playing different roles can interact and cooperate with each other and fulfill the tasks well under specific instruction.

## 4.3 Ablation Study

In this section, we eliminate generator, reranker, and evaluator, respectively, and explore to what extent the three aspects of LLMs' capabilities have an impact on the ODQA performance. In addition, we validate the effectiveness of the proposed prompt optimization.

It can be confirmed from Table 2 that the generator role of LLMs has the most significant impact among the three different roles played by LLMs, which indicates that the query expansion can serve

**Table 2: Ablation results on TriviaQA, WebQ, and NQ datasets.**

| Method | TriviaQA | WebQ | NQ |
|---|---|---|---|
| Ours w/o expansions generator | 68.70 | 52.61 | 53.66 |
| Ours w/o documents reranker | 73.06 | 52.71 | 54.68 |
| Ours w/o candidates evaluator | 73.91 | 55.56 | 57.12 |
| Ours w/o prompt optimization | 73.60 | 54.82 | 56.68 |
| **Ours** | **76.62** | **57.15** | **57.56** |

| Initial Query Expansion Prompt | Optimized Query Expansion Prompt |
|---|---|
| **You are serving as a generator. To answer the given question more precisely, provide background information from Wikipedia as the query expansion to enrich the context.** | You are serving as a generator **to generate query expansion.** To answer the given question more precisely, provide background information from Wikipedia **or give the analysis of the question** as the query expansion to enrich the context. |

| Initial Document Reranking Prompt | Optimized Document Reranking Prompt |
|---|---|
| **You are serving as a reranker, you should rank the given documents according to the following rules: 1. The more relevant the document is to the question and expansion, the higher the score is. 2. The more informative the document is, the higher the score is. 3. The more possible the document may contain the answer to the question, the higher the score is. Please make sure you have make comprehensive understanding of the above rules and documents. You should think step by step and rank the documents above carefully according to the rules.** | You are serving as a reranker, you should rank the given documents according to the following rules: 1. The more relevant the document is to the question and expansion, the higher the score is. 2. The more informative the document is, the higher the score is. 3. The more **likely** the document may contain the answer to the question, the higher the score is. **Please carefully consider the relevance, informativeness, and likelihood of containing the answer when ranking the documents.** You should think step by step and rank the documents above carefully according to the rules. |

**Figure 3: Case study for prompt optimization. EM score for initial prompt is 54.82; EM score for optimized prompt is 57.15. The result is reported on WebQ dataset.**

as an auxiliary document. The reranker contributes to the ODQA as well because the reranked the documents are more relevant to the question. The feasibility of evaluator has also been demonstrated as it can estimate the evidence quality and select the most suitable one. Our experiment on prompt optimization shows that the quality of the prompt design directly affects the performance of role-play LLMs for the final result and that the prompts on discrete space could be optimized as well.

## 4.4 Case Study and Error Analysis

**Case study of prompt optimization.** We analyse the differences of the prompts for query expansion and document reranking after optimization. From Figure 3 we can see that compared to the initial prompts, optimized prompts can include more details and insights for describing the instruction. As for the expansion prompt, a more detailed role-play description and an alternative instruction to solve the task were added. As for the reranking prompt, some of the ambiguous content in the prompt got refined after optimization. As a consequence, our proposed prompt optimization method is able to achieve more detailed, instructive and explicit prompts.

**Case study of evidence and answer.** In addition to prompt optimization, we also focus on the specific performance of evidence quality and answer generation during the inference process. We choose GenRead as a strong baseline for comparison. As shown in Table 3, the top-10 evidence documents of LLMQA all contains answers, which are highly relevant to the given question resulting in accurate answer prediction. However, the virtual documents generated by GenRead introduces an inaccurate year 1951 and misses the

**Table 3: Case study of more pertinent evidence than baseline.**

**Question:** when did little polveir win the grand national
**Golden Answer:** [1989]
*LLMQA*
**Selected Doc. Hit:** 10 / 10
**Top-3 docs:**
"...He won the 1989 Grand National steeplechase..."
"...on 8 April 1989. The race was won in a time..."
"...He is best known...for his performance in the 1989 Derby ..."
**Generated answer:** 1989 (True)
*GenRead*
**Selected Doc. Hit:** 1 / 10
**Top-3 docs:**
"On 6 April 2019, Little Polveir won the Grand ..."
"... Little Polveir won the Grand National in 1951."
"... last time Little Polveir won the Grand National was 1869."
**Generated answer:** 1951 (False)

**Table 4: Case study of imprecise evidence for hard example.**

**Question:** who wrote the first declaration of human rights
**Golden Answer:** [Cyrus]
*LLMQA*
**Selected Doc. Hit:** 2 / 10
**Top-3 docs:**
"...Cyrus the Great... the first human rights document..."
"...first recording of human rights ... by Cyrus the Great..."
"...is a human civil rights document ... the French Revolution."
**Generated answer:** Cyrus the Great (False)
*GenRead*
**Selected Doc. Hit:** 1/ 10
**Top-3 docs:**
"The first declaration of human...written by George Mason"
"...first declaration of human...Virginia Declaration..."
"...first declaration of human rights... Virginia Declaration..."
**Generated answer:** George Mason (False)

golden answer. This indicates that reranking of retrieved documents can help improve the evidence quality and answer accuracy.

**Error Analysis.** Although we achieve advanced results on evidence quality and answer accuracy, some questions remain challenging. These questions may contain contradictions with facts and world knowledge, and it may have led to incorrect predictions or reasoning results. For instance, Table 4 displays a typical failure case that both LLMQA and GenRead struggle to capture precise evidence, leading to low evidence quality and incorrect answer predictions. Despite these ongoing challenges, our LLMQA is still the best choice in terms of overall performance on the ODQA task.

## 4.5 Experimental Analysis

**Analysis of Evidence Quality.** We estimate the evidence quality using answer recall on the top@K selected documents. We compare our proposed LLMQA with GenRead [45] on three datasets. Table 5 shows that our LLMQA achieved highest recall on all of top@K settings over three datasets. The results show that relying solely on LLM-generated documents is insufficient. While hybrid

**Table 5: Answer recall for evidence quality.**

| Dataset | Method | Top@2 | Top@4 | Top@8 |
|---------|--------|-------|-------|-------|
| NQ | GenRead-sampling | 55.12 | 62.58 | 69.64 |
|  | GenRead-clustering | 55.12 | 62.58 | 69.64 |
|  | LLMQA | **61.99** | **78.59** | **82.94** |
| TriviaQA | GenRead-sampling | 73.55 | 77.99 | 81.55 |
|  | GenRead-clustering | 76.09 | 79.65 | 82.96 |
|  | LLMQA | **80.67** | **86.21** | **87.22** |
| WebQ | GenRead-sampling | 58.02 | 64.67 | 69.59 |
|  | GenRead-clustering | 61.17 | 67.47 | 72.00 |
|  | LLMQA | **67.57** | **77.81** | **80.07** |

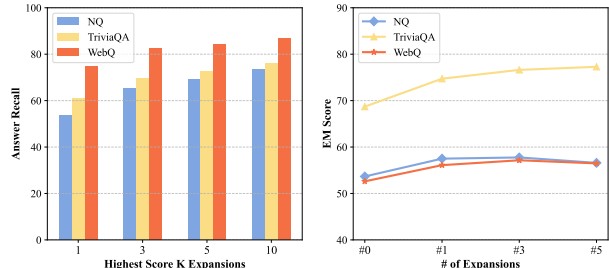

**Figure 4: Analysis of query expansion.**

utilization of both generated expansion and retrieved documents can gain tremendous answer recall increase, contributing to the final performance improvement.

**Quality of Query Expansions.** We first evaluate the quality of query expansions generated by LLMs. In the query expansion procedure, we generate 10 candidates and instruct LLMs to estimate the candidates according to the specified rules. The left in Figure 4 shows the recall for the highest score K expansions. Most of the generated query expansions have already contained the answer, because the large amount of knowledge that may cover the answer has been stored in the parameters of LLMs during pre-training.

We also analyse the number of expansions in our proposed approach. The right in Figure 4 shows the EM score for different numbers of expansions, the approach without expansion encounters massive performance drop and the approach with expansions can benefit from the increment of the expansions number. The result indicates that when the retrieved documents are of poor quality, the query expansions generated by the LLMs can be used as auxiliary documents to assist in selecting the relevant documents and answering the question.

As we treat the expansions as auxiliary documents, the location to insert the expansions may also have an impact. Constrained by the context length, expansions inserted to the beginning of the documents gain the best EM score, indicating that reader model may be much more sensitive to the beginning of the context.

**Strategies in Documents Reranking.** In order to demonstrate the reranking capability of LLMs, we implemented different strategies in reranking stage for document selection. As Table 6 shows, LLM-based sliding window reranking achieved the best result. Compared to using DPR score directly, sliding window reranking can have more comprehensive understanding of the retrieved documents and obtain the coarse-grain relevance between the question, expansion

**Table 6: Analysis of strategies in document selection.**

| Strategies | LLM Sliding | DPR Score | Random |
|------------|-------------|-----------|--------|
| NQ | **57.56** | 54.68 | 49.19 |
| TriviaQA | **76.62** | 73.04 | 69.42 |
| WebQ | **57.15** | 52.71 | 46.82 |

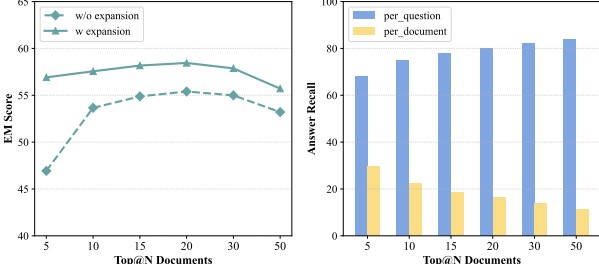

**Figure 5: Impact of document number.**

and documents, however the high-level embedding similarity used in DPR may include much noisy information.

**Impact of Document Number.** Intuitively, the question is more likely to be answered correctly when the number of retrieved documents is larger. We conducted an experiment on the number of documents input to the reader model, and Figure 5 shows that within a certain range, the EM score rises along with the number of documents, while when the number exceeds 30, the accuracy drops to some extend. This is because simply increasing the number of documents may lead to a decrease in the percentage of valid information as shown in Figure 5, making it difficult for the reader model to mine the correct answer from a large number of documents.

**Complexity Analysis.** Regarding the number of model parameters to be learned, we compare them from two aspects: evidence collection and answer generation. Some previous methods require specifying specialized models to collect evidence (*e.g.,* document retrieval and extension generation), which introduces the training cost for specialized models in evidence collection. Our framework is instead based on guiding LLMs to play different roles, with evidence collection only involving inference process. Since the inherent capabilities of the reader (answer generator) have an important impact on the performance of the ODQA task, state-of-the-art ODQA performance comes from fine-tuning the reader. Following [45], we employ T5-xl (3B) as the backbone of answer generator, whose training cost is comparable to the baseline.

## 5 CONCLUSION

In this paper, we propose LLMQA that formulate the ODQA generation process as three fundamental steps: query expansion, document selection, and answer generation, which combines the superiority of both retrieval-based and generation-based evidence. Since LLMs have showcased remarkable performance on generation, ranking and evaluation, we use a generalized framework to integrate multi role-play LLMs: generator, reranker and evaluator, which collaboratively contribute to each key step in the ODQA generation process. Furthermore, we design a novel prompt optimization algorithm, to address the limitation of prompt sensitivity, guiding LLMs in producing higher-quality evidence and more accurate answers.

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
