# OpenReview forum: "Harnessing Multi-role Capabilities of Large Language Models for Open-domain Question Answering"
_ACM.org/TheWebConf/2024/Conference — TheWebConf24 Oral_

### Official Review · Reviewer_CGcA · 2023-11-24

**Novelty:** 5
**Technical Quality:** 5

**Review:**

This paper proposes a multi-role approach with LLMs for open-domain QA. It is well-written and easy to follow. Many similar approaches have been discussed recently, and this paper provides good discussions and analysis for open-domain QA application scenarios. Overall, I recommend this paper for TheWebConf'24.

I have some suggestions and questions, and it would be great if we could get the answer in the revised version.

(1) To what extent will the "initial prompt" influence the performance? In other words, to what extent is the good performance caused by selecting a good initial prompt for each role? It is worth discussing with different paraphrased initial prompts.

(2) How's the performance if we use GPT-3.5, even 4.0, to play all roles? It may lead to various interesting discussions. For example, DPR vs. GPT-3.5 and P3 vs. GPT-3.5. Additionally, it can answer one more research question: whether we only need LLMs for this approach.

(3) Regarding Algorithm 1: How many times do we need for prompt optimization? When is a good time to stop the optimization process? Whether the more the better? Did you try other scoring (evaluating) functions?

**Questions:**

(1) To what extent will the "initial prompt" influence the performance?

(2) Regarding Algorithm 1: How many times do we need for prompt optimization? When is a good time to stop the optimization process?

**Reviewer Confidence:**

4: The reviewer is certain that the evaluation is correct and very familiar with the relevant literature

**Scope:**

4: The work is relevant to the Web and to the track, and is of broad interest to the community

---

### Official Review · Reviewer_db7Q · 2023-11-30

**Novelty:** 5
**Technical Quality:** 4

**Review:**

There are two dominant frameworks to model open-domain question answering:
1. retrieve and read
2. generate and read

The two paradigms above are inherently different in several ways as mentioned in the paper. What the authors do in this paper is that they learn these two paradigms jointly in a unified approach with the help of prompt optimisation. This is the key novelty of this work too.

While the work has advantages, there are some disadvantages too:
1. The performance of the model is hugely dependent on different components, while the paper has conducted an ablation analysis, what will be interesting to analyse is that one of the components does not perform ideally. The authors say that the model is a unified learning of the different components. From the figure, e.g., Figure 1, it is not actually unified learning. It is a cascaded model where the input of one phase goes as the output from the previous phase.

2. Generative information retrieval has been gaining a lot of attention in the recent past. It would be interesting to see in this paper how the model performs with traditional generative learning such as https://arxiv.org/abs/2202.06991.

**Questions:**

If the authors could address some of the key concerns in points 1 and 2, it would help me in significantly understanding this work.

**Ethics Review Description:**

NIL

**Reviewer Confidence:**

2: The reviewer is willing to defend the evaluation, but it is likely that the reviewer did not understand parts of the paper

**Scope:**

4: The work is relevant to the Web and to the track, and is of broad interest to the community

---

### Official Review · Reviewer_2Xkt · 2023-11-30

**Novelty:** 5
**Technical Quality:** 6

**Review:**

The paper proposes a generalized framework to combine retrieval and generation methods for better open-domain questions answering. It uses LLM to expand queries, rerank relevant documents and generate answers.

Pros:
The paper is well written and very clear, the storyline is clear and easy to follow.
The framework proposed is original and significant
The authors use some of the latest LLM
The authors propose a novel method for prompt optimization within this framework for OQDA which the authors show can greatly improve the performance of the method

Cons:
Technically the approach borrows from individual techniques and models already in literature but combines them to create novel framework that outperforms other state of the art methods for ODQA.

**Questions:**

For comparison it would also be useful to discuss the cost of your approach compared to existing approaches, since yours uses LLM in three different steps. It would also be very informative to provide time used in each of the three steps of your model (expansion, reranking and generation) at inference time.

You compare the reader parameters in the table, but you also use gpt for query expansion and reranking of documents, how many parameters has the LLM you used for these tasks?

Can you provide more information on the training process. In your framework the LLM for query expansion and document reranking is fixed, but you finetune the answer generation LLM and you optimize the prompts for query expansion, document reranking and answer generation. In which order do you do the trainings? Do you first finetune the answer generation LLM and then do prompt optimization?
In the implementation details might be useful to say how the generator is finetuned and which training data is used.

Can you provide also examples of initial prompt for answer generation and optimized prompt?

**Reviewer Confidence:**

4: The reviewer is certain that the evaluation is correct and very familiar with the relevant literature

**Scope:**

4: The work is relevant to the Web and to the track, and is of broad interest to the community

---

### Official Review · Reviewer_g2iy · 2023-12-01

**Novelty:** 6
**Technical Quality:** 5

**Review:**

# Summary
The authors introduce LLMQA, an approach to open-domain QA that uses LLMs to unify the existing retrieve-then-read and generate-then-read paradigms.  Concretely, different pretrained LLMs are prompted to a) expand questions with additional context; b) rerank retrieved documents; c) synthesize the query, expansion, and documents into the final answer; and d) evaluate candidate intermediate outputs.  To support this procedure, they further introduce a variational prompt optimization algorithm that jointly tunes the hard prompts used for expansion, reranking, and synthesizing.  They demonstrate that their method significantly outperforms retrieval-only and generation-only baselines on standard benchmark datasets.  The authors additionally show that LLMQA generates/selects higher-quality evidence.

# Strengths
* The proposed method effectively unifies two dominant paradigms and outperforms several different types of baselines.
* The prompt optimization algorithm is novel and contributes significant improvements to the overall method, as well as insights on prompt optimization techniques in general.
* The analyses and ablations provide insight into the building blocks of an effective ODQA system.

# Weaknesses
* The paper could benefit from clearer presentation of the methods and experimental setups, especially in the early parts of the paper.  Among other things, key details about the prompt optimization algorithm are not described.  See questions and organizational comments below.
* While the method is effective, it introduces a significant number of additional components and inference calls end-to-end.  Consequently, while the training complexity is only marginally more than that of the baselines, the complexity of evaluation is much greater.
* Some of the analyses could be expanded:
  * Query expansion quality (lines 840-862): it’s not clear that the claim in lines 852-853 holds unconditionally; there is a decrease from 3->5 expansions in two tasks, and additional insight on this would help to qualify this claim.
  * Document reranking strategies (lines 863-890): this would benefit from more thorough exploration of reranking strategies.  The other “strategies” presented are a random baseline and a retrieval-only baseline; while these are useful as ablation studies, they do not add further information on top of Table 2.

# Typos, style, organization
* Figure 2 (approx. line 361) – “expansoin”
* Algorithm 1, line 7: $s_{a_i}$ has not been defined
* Line 537: should the priors be subscripted with $\theta$ instead of $\phi$?
* Certain details could benefit from being introduced earlier in the paper.  Among other things:
  * Query expansion being used to generate documents, not to expand the query context for retrieval (in the style of doc2query [1]).
  * The presence of multiple candidates for query expansion and reranking, instead of just one, and the possibility of selecting more than one.
  * The different roles in LLMQA being filled by different LLMs, not one single LLM.

# References
[1] https://arxiv.org/abs/1904.08375

# Updates
1. I acknowledge that I have read the authors' rebuttals dated 09 Dec 2023.  I have provided follow-up comments.

**Questions:**

1. Section 3.2: Are the generated documents reranked with d or directly appended to the topK retrieved docs?  Section 4.5a suggests the former, while sections 3.4 and 4.5b suggest the latter.  If it is the former, what is generally the ratio of generated to retrieved docs?
2. Section 3.5: The two inputs to each of the equations (5) and (6) appear to be both sides of equations (2) and (3), respectively.  What is being operated on here?
3. Section 3.6: How are the initializations (for the models/prompts used to generate priors) created?
4. Section 3.6: What is the posterior generation process for query expansion?  There does not appear to be a notion corresponding to the “last document” used in reranking.
5. Section 3.6: What are the update functions?
6. Section 4.1: Please share the prompts for each component before and after optimization.  Did you control for prompt length during optimization?
7. Section 4.1: What is the process for converting one embedding into scalar scores/probabilities in [0, 1]?
8. Section 4.5, lines 860-862: Is this demonstrated in a data point?  It’s not clear that this follows from Figure 4.
9. Table 1: How many distinct LLMs, total available params, LLM calls, and trained parameters are used in each of these methods?
10. Table 2: Why are the Evaluators still required after tuning the prompt?  It would be expected that Expansion and Reranking with optimized prompts would have strong top1 candidates.

**Reviewer Confidence:**

3: The reviewer is confident but not certain that the evaluation is correct

**Scope:**

4: The work is relevant to the Web and to the track, and is of broad interest to the community

---

### Official Review · Reviewer_cUbK · 2023-12-05

**Novelty:** 5
**Technical Quality:** 5

**Review:**

This paper proposes an approach that exploits the multiple capabilities of LLM for different stages of ODQA. LLM is used to expand a question, to rank the retrieved documents and to evaluate the quality of them. Finally, LLM is asked to read the selected document to generate an answer. In addition, the prompts are optimized jointly. The experiments on 3 datasets show that the approach can improve the effectiveness of ODQA compared to the baselines.

The idea of instructing LLM for different subtasks of ODQA is very interesting. Its effectiveness is demonstrated in the experiments. The ablation study shows that each of the component contributes to the increase in effectiveness.

The paper also has several unclear aspects that should be addressed.

1. The prompt optimization step is not described. Instead, the authors refer to an existing paper for it. This hinders enormously the understanding of the paper. It is unclear what prompts are learned and resulted, and how the learning process can optimize them.
The form of prompts is not described in detail. There are some examples of the initial prompt and optimized prompt. It is understood that the prompts are in natural language. It is unclear how prompt optimization can learn an optimized discrete prompts. I confess that I have (purposely) not read the paper in reference. A research paper should be self-content, which is not the case here. As a central aspect of the paper, prompt optimization should be at least described to some extent.

2. Query expansion is one of the steps used. It is unclear what such a process will produce. Is it an enhanced search query produced by a prompt to LLM? Is it a set of texts generated by LLM and retrieved that serve for the purpose of expansion? In the latter case, how are the documents incorporated into query expansion?

Similar questions arise: the description uses model parameters to characterize each step, while natural language prompts are shown in the figures and examples. It is unclear how they match.

3. In table 1, statistical significance is indicated. What is the test performed? Is the performance compared to all the others, or the strongest competitor?

4. The window-based reranking may be an interesting idea, but it is unclear to me why this may be better than a global reranking process that considers all the candidates together.

5. What is the reason to have both reranker and reranking evaluator? Would it be possible or simpler to combine them into a single process?

**Questions:**

See comments above.

**Reviewer Confidence:**

3: The reviewer is confident but not certain that the evaluation is correct

**Scope:**

4: The work is relevant to the Web and to the track, and is of broad interest to the community

---

### Decision · Program_Chairs · 2024-01-22

**Decision:**

Accept (Oral)

**Comment:**

This paper presents a novel approach to open-domain question answering (ODQA) that leverages large language models (LLMs) for different roles, such as query expansion, document reranking, answer generation, and evaluation. The paper also introduces a variational prompt optimization algorithm that jointly tunes the hard prompts used to instruct the LLMs for each role. The paper shows significant improvements over existing retrieval-only and generation-only baselines on standard benchmarks, as well as higher-quality evidence selection.

 The work is original, has strong experimental results and the prompt optimization method is quite novel. The paper is well written. The analysis and ablations give a nice insight into the building blocks of a ODQA system.

 The reviewers identified a few weaknesses, which the authors have addressed largely to their satisfaction.

 I recommend an accept with recommended revisions. Although the paper has some weaknesses that need to be clarified or addressed in a revision, it has strong merits that outweigh them.